# Self-report and device-based physical activity measures and adherence to physical activity recommendations: a cross-sectional survey among people with inflammatory joint disease in four European countries

N Brodin,[1,2] David Moulaee Conradsson,[1,3] Thijs Willem Swinnen ![ORCID] ,[4,5] Bente Appel Esbensen,[6,7] Norelee Kennedy ![ORCID] ,[8,9] Nanna Maria Hammer,[6] Sean McKenna,[10] Peter Henriksson,[11] Birgitta Nordgren ![ORCID] [1,3]

For numbered affiliations see end of article.

**Correspondence to**
Dr Birgitta Nordgren;
Birgitta.Nordgren@ki.se

## ABSTRACT

**Objectives** Self-monitoring of physical activity (PA) has the potential to contribute to successful behaviour change in PA interventions in different populations, including people with inflammatory joint diseases (IJDs). The objectives of this study were to describe the use and knowledge of self-report-based and device-based PA measures in people with IJDs in four European countries, and to explore if the use of such devices, sociodemographic or disease-related variables were associated with adherence to the recommendations of at least 150 min of moderate to vigorous PA per week.

**Setting** Cross-sectional survey, performed in 2015–2016.

**Participants** People with IJDs in Belgium, Denmark, Ireland and Sweden.

**Primary and secondary outcome measures** Use of self-report and device-based PA measures, receipt of instructions how to use PA measures, confidence in using them, adherence to PA recommendations and associated factors for adherence to PA recommendations.

**Results** Of the 1305 respondents answering questions on PA measures, 600 (46%) reported use of any kind of self-report or device-based measures to self-monitor PA. Between country differences of 34%–58% was observed. Six per cent and four per cent received instructions from health professionals on how to use simple and complex devices, respectively. Independent associated factors of fulfilment of recommendations of PA were living in Ireland (OR=84.89, p<0.001) and Sweden (OR=1.68, p=0.017) compared with living in Denmark, not perceiving activity limitations in moderate activities (OR=1.92, p<0.001) and using a device to measure PA (OR=1.56, p<0.001). Those living in Belgium (OR=0.21, p<0.001) were less likely to fulfil recommendations of PA.

**Conclusions** Almost half of the participants with IJDs used self-report-based or deviced-based PA measures, although few used wearable devices regularly. The results indicate that participants meeting public PA health guidelines were engaged in self-monitoring of PA.

## STRENGTHS AND LIMITATIONS OF THIS STUDY

⇒ First study describing the use of self-report-based and/or device-based physical activity (PA) measures in people with inflammatory joint disease in a real-world setting across four European countries.

⇒ Study participants had low confidence in using device-based PA measures.

⇒ Study participants might have been particularly interested in PA prompting their participation in the study, limiting external validity.

⇒ Due to the cross-sectional design, conclusions on causal relationships cannot be drawn.

## INTRODUCTION

Regular physical activity (PA) is associated with lower risk of cardiovascular disease, diabetes and several cancers. Healthy behaviours such as achieving sufficient PA is a challenge for most people but may be even more challenging for people with inflammatory joint diseases (IJDs), as IJDs are characterised by functional limitations, pain and fatigue.[1 2]

PA defined as 'any bodily movement produced by skeletal muscles that result in energy expenditure'[3] is safe and can improve disease activity, pain, fatigue, quality of life and sleep in people with IJDs.[4 5] Despite these beneficial health outcomes, people with IJDs have lower PA levels than their healthier counterparts and many do not meet the required PA recommendations.[6–10]

Self-monitoring of PA has the potential to contribute to successful behaviour change in PA interventions in different populations, including people with IJDs.[11–14] PA can be monitored in different ways, although

measurement properties (ie, validity, reliability and responsiveness to change) vary considerably across techniques. Common methods for monitoring PA range from simple questionnaires and pedometers to more complex devices such as accelerometers.[15] Self-reported measures such as diaries and questionnaires as well as device-based PA measures can raise awareness and motivate patients to initiate and maintain PA.[12] In particular, device-based wearable trackers can give the user valid data on, for example, daily steps or PA intensity levels.[16] Given their superior feasibility, adequate reliability and sufficient construct validity in group analyses, self-reported PA questionnaires remain useful to relate PA behaviours to clinical outcome in larger arthritis research studies.[17 18] Moreover, health professionals (HPs) play an important role in supporting PA behaviour in people with IJDs and could contribute to better health outcomes through promoting PA monitoring in clinical practice.[19–21] Better knowledge and understanding of people with IJDs use and awareness of PA measures and its association to PA levels is needed to guide future practice of health promotion in this population. The use of PA measures in different European countries and to what extent people with IJDs perceive it to be important in measuring PA behaviour needs to be further explored.

The main objective of this study was therefore to describe the use, knowledge and sources of information of self-report-based and device-based PA measures in people with IJD in four European countries. A further objective was to explore if the use of such devices, sociodemographic factors or disease-related variables were associated with adherence to the recommendations of at least 150 min of moderate to vigorous PA per week.

## METHODS
### Design
Cross-sectional design using questionnaire was chosen as it allowed for the collection of data from a wider range of participants. The Strengthening the Reporting of Observational Studies in Epidemiology reporting guidelines for observational studies were used to guide the reporting of this study.[22]

### Patient and public involvement
Patients representing different IJDs in the four countries were involved in the design of the questionnaire. This was organised following the same structure in all countries. The patients were invited by each country representative to give their views and input on the questionnaire. They were asked to comment on the content of the questionnaire and identify any missing or problematic questions/constructs. All input was considered in the final questionnaire.

### Sample
People with IJDs in four European countries, that is, Belgium, Denmark, Ireland and Sweden were invited to

participate in the study. In Belgium, people were identified through an outpatient university clinic and through patient organisations. In Denmark and Ireland, people were identified through the country's patient organisation membership. In Sweden, recruitment was performed through a Facebook group hosted by the patient organisation and from an outpatient rheumatology clinic. The chairperson for each country's patient organisation was contacted, requesting permission for their members to participate in the study. Following the granted permission, the chairpersons in each country received information via email, containing study information, survey link and researcher details and they provided the members with the same. In Sweden, participants recruited at the rheumatology clinic received the same information mentioned above as well as the paper questionnaire distributed through the patient organisation. Consent to participate was provided by responding to the survey anonymously.

### Questionnaire
The study steering group developed a questionnaire divided into four domains: (1) sociodemographic; (2) disease-related factors, (3) PA self-monitoring measures; questionnaire and diary (paper or digital), simple body worn PA devices, (eg, simple wearable activity trackers) or complex body worn devices (eg, accelerometer and wearable activity trackers with wireless link to smartphone app or website) and (4) questions about receiving instructions how to use PA measures, the importance of measuring PA, confidence in using PA measures; and if having been recommended by an HP to self-monitor PA using any self-report or device-derived measure (online supplemental file).

The questionnaire was developed in English and translated into each country's language. To ascertain face validity of each country's version of the questionnaire, discussions were organised with HPs, that is, physiotherapist, occupational therapist and a clinical nurse specialist. In addition, interviews with four people diagnosed with different IJD (rheumatoid arthritis (RA) n=2, ankylosing spondylitis (AS) n=1 and psoriatic arthritis (PSA) n=1)) to cover different disease spectra were also organised countrywise. These discussions took place to explore whether the constructs surveyed within each questionnaire reflected the aims under study (ie, to identify missing or problematic questions/constructs) and if the questions were well understood. We; therefore, applied a purposive sampling method to reflect different diagnoses of arthritis, age and gender and subsequently the questionnaires were adapted to improve their readability, validity and specificity. The final questionnaires were piloted with people with IJD and experts in this field to ensure content validity. Thus, verifying whether questions were readable, relevant and representative of the study's aims, in addition to minimise the risk of missing data.

PA levels were measured using the The Short Questionnaire to Assess Health Enhancing Physical Activity

(SQUASH) questionnaire.[23] The SQUASH allows a comparison of national and international PA recommendations and includes questions on commuting activities (walking, cycling), PA at work or school, household activities and leisure time activities (walking, gardening, cycling, sports). The participants were asked to refer to a normal week during the last month; how many days a week they engaged in each of the activities, the average time per day spent on each activity (hours and minutes) and the effort of each activity (low, moderate, high). Activities were assigned a Metabolic Equivalent Task value according to Ainsworth's' compendium of physical activities.[24] Based on reported effort of the activities in the questionnaire, activities were assigned an intensity score between 1 and 9, were ≥3 assumed to represent health-enhancing PA. Study participants, who reported 150 min or more of moderate or more intensive activities per week, were classified as reaching health-enhancing PA recommendations. This was calculated by summing up the number of days per week for activities on a moderate or higher intensity level where the total duration was 30 min or more. A minimum of 5 days resulted in patients being categorised as reaching the health-enhancing PA recommendations.[5] The SQUASH has previously demonstrated good test–retest reliability and modest construct validity in the IJD population.[17]

### Data collection

The survey was conducted online during 2015–2016 with one reminder sent. For participants in Belgium, Denmark and Ireland, the survey was distributed through SurveyMonkey or SurveyXact. In Sweden, the survey was conducted online through Artologik Survey&Report as well as through paper questionnaire distributed at an outpatient rheumatology clinic.

### Analysis

Statistical analyses were carried out using IBM SPSS, V.27.0 (SPSS).

For the main objective, the results were presented descriptively using proportions and percentage or median and IQR for categorical variables and means and SD for continuous variables.

For further objectives, table 1 displays variables and criteria used for categorisation of independent and dependent variables for logistic regressions.

First, potential-associated factors for fulfilment of PA recommendation were entered one at a time in unadjusted univariate logistic regression models. At this point, the cut-off to include factors in the multivariate analyses was set to an alpha level p≤0.1. Prior to multivariable analyses, measures of association, that is, $\chi^2$ tests, were computed to assess for collinearity among potential associated factors. Associated factors were subsequently entered into a multivariable logistic regression model with an alpha level set at p≤0.05 for the identification of independent associated factors. Results for logistic regressions are expressed as OR and 95% CI.

**Table 1** Variables and criteria used for categorisation of the independent variables and the dependent variable

| Variables and instruments | Criteria for categorisation |
|---|---|
| Independent variables | |
| Country | Belgium/Denmark/Ireland/Sweden// |
| Age | ≥55 years/<55 years |
| Sex | Female/male |
| Work status | Working/not working |
| Education* | Higher/basic |
| Bio-DMARDs medication Non-bio medication | Treatment/no treatment Treatment/no treatment |
| Activity limitation—moderate activities | No limitation/limitations |
| Activity limitation—stairs | No limitation/limitations |
| Pain arm | No pain/pain |
| Pain hand | No pain/pain |
| Pain foot | No pain/pain |
| Pain leg | No pain/pain |
| Think it is important to monitor physical activity | Yes/no |
| Use device measuring physical activity | Yes/no |
| Dependent variable | |
| ≥150 min a week of moderate–high intensity physical activity | Yes/no |

*Higher education=university level. Basic education=primary and secondary level
DMARDs, disease-modifying antirheumatic drugs.

## RESULTS
### Study participants

A total of 1382 participants with IJD in Belgium, Denmark, Ireland and Sweden answered the survey mainly online. A majority of the participants were diagnosed with RA (n=554, 40%), 331 (24%) with AS, 296 (21%) with PSA and 201 (15%) with other diagnoses (table 2).

Descriptive data on demographic and disease-related factors and fulfilment of PA recommendations of participants is displayed in table 2.

### Use of PA measures

Of the included 1382 participants, 1305 answered questions on PA measures. Of those, 600 (46%) reported that they used any kind of self-report-based or device-based measures to self-monitor PA. The most common used measure was the simple body worn device, used by 428 (33%), whereof 112 used it regularly. The complex body worn device were used by 182 (14%), whereof 56 used it regularly. One hundred and seven of 1305 (8%) reported using digital or paper questionnaire and 104 (8%) used digital or paper diaries for self-monitoring of PA. Divided per country, the use of any kind of self-report-based or device-based measure differed between countries from 34% to 58% (table 2).

**Table 2** Demographic and disease-related factors and fulfilment of PA recommendations of participants

| Variables | Belgium (n=591) | Denmark (n=579) | Ireland (n=90) | Sweden (n=122) | Total (n=1382) |
|---|---|---|---|---|---|
| Diagnosis n (%) | | | | | |
| RA | 217 (37) | 237 (41) | 56 (62) | 44 (36) | 554 (40) |
| AS | 221 (37) | 77 (13) | 13 (14) | 20 (16) | 331 (24) |
| PSA | 101 (17) | 162 (28) | 15 (17) | 18 (15) | 296 (21) |
| Other* | 52 (9) | 103 (18) | 6 (7) | 40 (33) | 201 (15) |
| Female sex n (%) | 409 (69) | 509 (88) | 69 (77) | 108 (86) | 1095 (79) |
| Age, <55 years (%) | 263 (45) | 347 (60) | 48 (53) | 48 (39) | 706 (51) |
| Years symptom, mean (SD) | 17.9 (12.4) | 13.7 (10.8) | 10.1 (7.9) | 15.2 (12.2) | 15.4 (11.8) |
| Bio-DMARDs medication (%) | 150 (25) | 136 (24) | 39 (43) | 48 (40) | 373 (27) |
| Non-biological medication n (%) | 105 (18) | 306 (53) | 26 (29) | 39 (32) | 476 (34) |
| Basic education n (%)† | 289 (49) | 259 (45) | 37 (41) | 60 (49) | 645 (48) |
| Working full/part time n (%) | 203 (34) | 234 (40) | 44 (49) | 70 (57) | 551 (40) |
| Pain | | | | | |
| Hand n (%) | 285 (48) | 371 (64) | 40 (44) | 81 (66) | 777 (56) |
| Arm n (%) | 149 (25) | 327 (57) | 50 (56) | 79 (65) | 605 (44) |
| Foot n (%) | 248 (42) | 306 (53) | 31 (34) | 69 (57) | 654 (47) |
| Leg n (%) | 135 (23) | 335 (58) | 43 (48) | 95 (79) | 608 (44) |
| Back n (%) | 323 (55) | 307 (53) | 18 (20) | 66 (54) | 714 (52) |
| Neck n (%) | 271 (46) | 219 (38) | 15 (17) | 49 (40) | 554 (40) |
| Head n (%) | 80 (14) | 55 (9) | 2 (2) | 12 (9) | 149 (11) |
| Activity limitation | | | | | |
| Limited a lot moderate activities n (%) | 113 (19) | 156 (27) | 21 (23) | 17 (14) | 307 (22) |
| Limited a lot climbing stairs n (%) | 60 (10) | 147 (25) | 26 (29) | 20 (16) | 253 (18) |
| Own or use PA device n (%) | 340 (58) | 274 (47) | 49 (54) | 42 (34) | 600 (43) |
| Fulfilment of PA recommendations n (%) | 84 (14) | 277 (48) | 68 (76) | 74 (61) | 503 (36) |

*Other = fibromyalgia, lupus, systemic lupus erythematosus, Wegeners granulomatosis, psoriasis, scleroderma, arthritis.
†Basic education=primary and secondary level.
AS, ankylosing spondylitis; DMARDs, disease-modifying antirheumatic drugs; PA, physical activity; PSA, psoriatic arthritis; RA, rheumatoid arthritis.

### Instructions of PA measures by HPs

Twenty-six of the 428 (6%) users of simple devices received instructions from HPs on how to use the device and of those 182 using the complex device, 8 (4%) did. HPs instructed 52 (50%) of those 107 using digital/paper questionnaire and 10 out of 104 (10%) of those using diaries.

### Perceived importance and confidence in self-monitoring of PA

All the 1382 answered questions about importance and confidence. Most of the participants, 925 (67%) perceived it was important to measure PA and 558 (40%) reported they did not receive information about PA from any source. Of those receiving information about PA, 388 (28%) received the main source of information from a physiotherapist, 363 (26%) from a rheumatologist, 299 (22%) from internet, 159 (12%) from a general practitioner and 147 (11%) from newspapers or magazines. A small number of the study participants received

information from friends, 59 (4%), from an occupational therapist 46 (3%) or from a nurse 38 (3%).

The participants were most confident in using and interpreting PA data from simple body worn devices and paper questionnaires. They were least confident in using complex body worn sensors collecting multiple data, as well as using digital diaries (table 3).

### Fulfilment of PA recommendations and its associated factors

A total of 503 of the 1382 (36%) participants reported they fulfilled the recommendations of ≥150 min of PA on a moderate intensity per week, whereof 418 (39%) were female.

Univariate logistic regression analyses (table 4) revealed that those who lived in Ireland (OR=74.14, p<0.001) and Sweden (OR=1.68, p<0.001) were more likely to fulfil the recommendations of PA as compared with those who lived in Denmark, whereas those who lived in Belgium (OR=0.18, p<0.001) were less likely to fulfil the

**Table 3** Participants confidence/familiarity in using self-reports and devices based physical activity measures (n=1305)

|  | n | Median (IQR) |
| --- | --- | --- |
| Simple (0–10)* | 1305 | 3 (8) |
| Complex sensor (0–10) | 1305 | 1 (6) |
| Paper QR (0–10) | 1305 | 3 (5) |
| Paper diary (0–10) | 1304 | 1 (5) |
| Digital QR (0–10) | 1305 | 1 (5) |
| Digital diary (0–10) | 1305 | 0 (5) |

*Possible score range 0–10 0=not confident, 10=very confident.

recommendations of PA. The results also showed that participants who were <55 years of age (OR=1.46, p=0.001), working part time or full time (OR=1.61, p<0.001), being treated with non-biological disease-modifying medication (OR=1.54, p<0.001), with no perceived activity limitations in moderate activities (OR=1.41, p<0.001) and who were using a device to measure PA (OR=1.93, p<0.001) were more likely to fulfil the recommendations of PA. In contrast, those of male sex (OR=0.68, p=0.007) and not experiencing pain in the arm (OR=0.56, p<0.001) or leg (OR=0.65, p<0.001) were less likely to fulfil the recommendations of PA.

The multivariable logistic regression model (table 4) showed that participants who lived in Ireland (OR=84.89, p<0.001) and Sweden (OR=1.68, p=0.017) were more likely to fulfil the recommendations of PA as compared with those who lived in Denmark, whereas those who lived in Belgium (OR=0.21, p<0.001) were less likely to fulfil the recommendations of PA . Furthermore, participants not perceiving activity limitations in moderate activities (OR=1.92, p<0.001), and using a device to measure PA (OR=1.56, p<0.001) remained independent associated factors of fulfilment of recommendations of PA. In this model, 72.5% of the outcome was correctly classified.

## DISCUSSION

To the best of our knowledge, this is the first study describing the use of self-report-based and/or device-based PA measures in individuals with IJD in a real-world situation, across four European countries and its association to fulfilment of PA recommendations.

Overall, close to 50% of the participants used any kind of self-report-based or device-based measures to self-monitor PA. Furthermore, in our cohort 37% used device-based measures, which is comparable to the general population at the current time.[25] This is promising given the motivation for PA which comes with the use of devices and also their advantages of simplicity and accuracy.[25 26] A between country difference of 34%–58% was observed. It can be considered obvious that younger participants to a larger extent would be those who use wearable trackers. This could be considered supported by the data from

Sweden, which has the lowest amount of tracker users, and also is the country with the fewest participants under the age of 55 years. However, Belgium had the highest number of tracker users although the country did not have the youngest group. Hence, other explanations to the between country differences may be sought beyond the variables measured in this study.

Only 112 (8%) of the participants used simple body worn devices and 56 (4%) used complex body worn devices regularly. This is in line with previous interventions aiming to increase PA using wearable trackers in people with rheumatic and musculoskeletal diseases, showing that adherence to use of trackers over longer periods of time is not maintained.[25]

Although a majority (67%) of our sample perceived it was important to measure PA, 40% did not receive information from an HP on why it is important, which is similar to an earlier study in an arthritis population.[21] Of those using simple and complex devices only 6% and 4%, respectively, received instructions on how to use the devices. On the other hand, a substantial part, 50%, reported they were instructed how to use questionnaires to measure PA. One explanation to this could be that HPs are used to using traditional questionnaires in clinic and therefore might be more prone to inform their patients about them. Since our participants responded to our survey in 2015–2016, measurement of PA has most likely become more common in clinical practice as well as HPs knowledge about different methods available. One recent study suggest that HPs may be more experienced in using PA devices today.[27] The factor most strongly associated with engaging in PA is getting recommendation from an HP and previous research has identified low HP knowledge about PA and exercise as a reason for not guiding patients with IJD in PA.[28] Even though HPs are aware of benefits and barriers in relation to PA in IJD, promotion and advice is conflicted by HPs lack of certainty with respect to joint damage and fear of flare-up.[29] Moreover, recent studies highlight further educational needs and development for HPs to support PA in this population.[19 30] While PA devices can give the user real-time feedback of daily steps or energy expenditure, patients with IJDs would most likely benefit from also adding HPs support providing them with tailored feedback to motivate PA.[1] In fact, better long-term adherence has been shown in interventions incorporating feedback and behavioural change techniques compared with activity devices alone.[31]

The participants' confidence in using self-reports and devices was low. This highlights the importance of HPs communicating with patients on the importance of PA measurement devices to build confidence in their use and their role in changing behaviour to improve PA levels. With HPs increased experience of using device-based measures,[27] there is a possibility that communication to patients might increase.

Both univariate and multivariate regression models showed that participants living in Ireland and Sweden were more likely to fulfil the recommendations of PA as

**Table 4** Univariate and multiple logistic regression models for factors associated with fulfilment of recommendations of physical activity

| Variable | Univariate regression | | Multiple regression | |
|---|---|---|---|---|
| | OR (95% CI) | P value | OR (95% CI) | P value |
| **Country** | | | | |
| Belgium | 0.18 (0.14 to 0.24) | <0.001 | 0.21 (0.15 to 0.29) | <0.001 |
| Ireland | 74.14 (10.22 to 537.52) | <0.001 | 84.89 (11.64 to 619.23) | <0.001 |
| Sweden | 1.68 (1.13 to 2.50) | <0.001 | 1.68 (1.10 to 2.56) | 0.017 |
| Denmark | 1.00 (reference) | | 1.00 (reference) | |
| **Age** | | | | |
| <55 years | 1.46 (1.74 to 1.83) | 0.001 | 1.28 (0.97 to 1.68) | 0.076 |
| ≥55 years | 1.00 (reference) | | 1.00 (reference) | |
| **Sex** | | | | |
| Male | 0.68 (0.51 to 0.90) | 0.007 | 0.99 (0.70 to 1.42) | 0.971 |
| Female | 1.00 (reference) | | 1.00 (reference) | |
| **Work status** | | | | |
| Working | 1.61 (1.28 to 2.01) | <0.001 | 1.41 (0.87 to 1.50) | 0.565 |
| Not working | 1.00 (reference) | | 1.00 (reference) | |
| **Education** | | | | |
| Higher education* | 0.89 (0.81 to 1.27) | 0.895 | | |
| Basic education† | 1.00 (reference) | | | |
| **Bio-DMARDs medication** | | | | |
| Treatment | 1.24 (0.97 to 1.60) | 0.084 | | |
| No treatment | 1.00 (reference) | | | |
| **Non-biological medication** | | | | |
| Treatment | 1.54 (1.22 to 1.94) | <0.001 | 0.92 (0.70 to 1.22) | 0.565 |
| No treatment | 1.00 (reference) | | 1.00 (reference) | |
| **Activity limitation—moderate activities** | | | | |
| No limitation | 1.41 (1.09 to 1.83) | <0.001 | 1.92 (1.38 to 2.66) | <0.001 |
| Limitations | 1.00 (reference) | | 1.00 (reference) | |
| **Activity limitation—climbing stairs** | | | | |
| No limitation | 1.00 (0.79 to 1.27) | 0.990 | | |
| Limitations | 1.00 (reference) | | | |
| **Pain arm** | | | | |
| No pain | 0.56 (0.44 to 0.69) | <0.001 | 0.76 (0.57 to 1.01) | 0.059 |
| Pain | 1.00 (reference) | | 1.00 (reference) | |
| **Pain hand** | | | | |
| No pain | 0.89 (0.72 to 1.12) | 0.328 | | |
| Pain | 1.00 (reference) | | | |
| **Pain foot** | | | | |
| No pain | 1.10 (0.88 to 1.37) | 0.407 | | |
| Pain | 1.00 (reference) | | | |
| **Pain leg** | | | | |
| No pain | 0.65 (0.52 to 0.81) | <0.001 | 1.24 (0.92 to 1.66) | 0.153 |
| Pain | 1.00 (reference) | | 1.00 (reference) | |
| **Think it is important to monitor physical activity** | | | | |
| Yes | 1.26 (0.99 to 1.61) | 0.058 | | |

Continued

**Table 4** Continued

| Variable | Univariate regression | | Multiple regression | |
|---|---|---|---|---|
| | OR (95% CI) | P value | OR (95% CI) | P value |
| No | 1.00 (reference) | | | |
| Use device to measure physical activity | | | | |
| Yes | 1.93 (1.53 to 2.42) | <0.001 | 1.56 (1.20 to 2.026) | <0.001 |
| No | 1.00 (reference) | | 1.00 (reference) | |

*Higher education=university level.
†Basic education=primary and secondary level.
DMARDs, disease-modifying antirheumatic drugs.

compared with the participants who lived in Denmark, whereas those in Belgium were less likely to fulfil the PA recommendation. This is in line with other studies describing PA levels in these countries.[32–34] Furthermore, using a PA device was in addition an independent associated factor of adherence to PA recommendations, also when considering sociodemographic-related and disease-related factors, support from HPs and participants own perception of familiarity and confidence of using PA devices. As this is a cross-sectional study, we cannot determine if the measure itself motivates to PA in this sample, or if the more physically active participants use the measures for self-monitoring of their behaviour. However, using PA devices can be a valuable way to support people with IJDs to improve motivation or PA levels, and it is suggested to be taken into account in the promotion of PA.[25 35 36] Not perceiving activity limitations was also a factor associated to adherence to PA recommendations in our sample . This corresponds well with a previous review by Veldhuijzen van Zanten *et al*, where activity limitation is described as a barrier of PA.[1] These factors provide information for clinicians to identify which patients need more targeted support to improve their PA levels, an important consideration in managing busy clinical loads.

## LIMITATIONS

First, the study had a cross-sectional design therefore, conclusions on causal relationships cannot be drawn. This could have been helped by a follow-up over time to allow for firmer conclusions about causality.

Second, the questionnaire was based on self-report through social networking, which may have attracted socially desired answers. This is an inherent limitation in most surveys and while it was ensured the data collection was anonymous, self-reports are never free from bias.[37] Moreover, most of those who answered were those involved in the patient organisation, probably with a specific interest in their arthritis, and most likely had a particular interest in the topic. However, as inclusion also took place through outpatient clinics in both Belgium and Sweden, our sample is probably representative of a broader range of characteristics as the patient organisations' members are generally older,

thus not working and not on disability leave as they are retired. It should be noted that the overall response rate could not be calculated, as exact numbers of members of the patient organisations were not available or could not be retrieved. Another point which could be regarded as a limitation is that the recruitment procedure through the patient organisations was not similar in each country. In Belgium, Denmark and Ireland, invitations to participate were sent via email to members of the patient organisations while in Sweden recruitment took place through the Swedish Rheumatism Association's Facebook page. Furthermore, PA interventions using wearable trackers in people with IJDs have increased the past years.[38 39] This may have promoted increased awareness and use of PA devices since our data was collected. However, we are not aware of other studies which have addressed the same issues as we have. Our data should, therefore, remain relevant as a starting point for further studies.

As reported earlier in our previous study, investigating rheumatology HPs awareness of PA measures in people with IJDs[30] we cannot neglect the risk of respondents having misinterpreted questions in the survey. Simultaneously with the survey used in the prior study[30] the present survey was designed in English and translated into Swedish, Danish, French and Flemish with results back-translated into English for the reporting. Through the different translations, there is a risk that valuable information was lost. Face validity was investigated in each country before the survey was used, and part of the included questions were other validated questionnaires.[23 40]

## CONCLUSION

Despite these limitations, this study contributes and provides insight into the use of PA measures and devices in a real-world setting by people with IJDs. This is important in order to enhance the collaborative promotion of PA between HP and patient.[31] The study found that almost half of the participants with IJDs used self-based or deviced-based PA measures, signalling the value of such devices in improving overall PA awareness and engagement. Moreover, participants meeting

public PA health guidelines were engaged in self-monitoring PA. The study also identified factors associated with PA device use allowing for more targeted management of people with IJDs to improve PA. It also highlights the need for HP engagement with PA advice and understanding of PA devices as an important part of the care of people with IJDs.

**Author affiliations**
[1]Department of Neurobiology Care Sciences and Society, Division of Physiotherapy, Karolinska Institutet, Stockholm, Sweden
[2]Division of Physiotherapy, Orthopaedic Clinic, Danderyds Sjukhus AB, Stockholm, Sweden
[3]Medical Unit Occupational Therapy & Physiotherapy, Theme Women's Health and Allied Health Professional, Karolinska University Hospital, Stockholm, Sweden
[4]Department of Rehabilitation Sciences, KU Leuven, Leuven, Belgium
[5]Division of Rheumatology, KU Leuven University Hospitals Leuven, Leuven, Belgium
[6]Copenhagen Center for Arthritis Research (Copecare), Copenhagen University Hospital - Rigshospitalet, Glostrup, Denmark
[7]Department of Clinical Medicine, University of Copenhagen, Kobenhavn, Denmark
[8]Discipline of Physiotherapy, School of Allied Health, University of Limerick, Limerick, Ireland
[9]Health Research Institute, University of Limerick, Limerick, Ireland
[10]Health Service Executive, Department of Physiotherapy, University of Limerick Hospitals Group, Dooradoyle, Ireland
[11]Department of Clinical Sciences, Danderyds Hospital, Karolinska Institutet, Stockholm, Sweden

**Acknowledgements** The authors thank Sara Wiljeminis who assisted in designing the survey and collecting the data.

**Contributors** NB, BN, NK, BAE and TS were the original authors who submitted the grant, designed the study, oversaw the data collection and analysis in their countries, prepared the paper and read and commented on all drafts and agreed on the final manuscript. SM and NMH were research assistants who assisted with designing the survey, collecting the data and commented on manuscript drafts. DMC and PH undertook the statistical analyses and commented on all manuscript drafts. BN is responsible for the overall content as guarantor and accepts full responsibility for the finished work and/or the conduct of the study, had access to the data, and controlled the decision to publish.

**Funding** The study was funded by the European Leauge Against Rheumatism Health Professional Research Grant 2015.

**Competing interests** None declared.

**Patient and public involvement** Patients and/or the public were involved in the design, or conduct, or reporting, or dissemination plans of this research. Refer to the Methods section for further details.

**Patient consent for publication** Not applicable.

**Ethics approval** This study involves human participants and was approved by Medical Ethical Committee of the University Hospitals Leuven S58849, Ethics Committee of the Capital Region of Denmark H-15013386, University of Limerick research ethics committee 2015_09_02_EHS, The Swedish Etikprövningsmyndigheten 2015/1785-31/5. Participants gave informed consent to participate in the study before taking part.

**Provenance and peer review** Not commissioned; externally peer reviewed.

**Data availability statement** No data are available.

**ORCID iDs**
Thijs Willem Swinnen http://orcid.org/0000-0002-5289-1023
Norelee Kennedy http://orcid.org/0000-0001-6047-1240
Birgitta Nordgren http://orcid.org/0000-0001-8903-1273

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
