## [Reviewer comments · BMJ Open]

ARTICLE DETAILS

TITLE (PROVISIONAL)	Self-report and device-based physical activity measures and adherence to physical activity recommendations. A cross-sectional survey among people with inflammatory joint disease in four European countries.
AUTHORS	Brodin, N; Conradsson, David; Swinnen, Thijs; Esbensen, Bente; Kennedy, Norelee; Hammer, Nanna Maria; McKenna, Sean; Henriksson, Peter; Nordgren, Birgitta

VERSION 1 – REVIEW

REVIEWER	Davergne, Thomas Sorbonne Université, Institut Pierre Louis d'Epidémiologie
REVIEW RETURNED	24-Jul-2022

GENERAL COMMENTS	I would like to thank the editor for allowing me to review the article “Self-report and device-based physical activity measures and their contribution to adherence to physical activity recommendations. A cross-sectional survey among people with inflammatory joint disease in four European countries. “ in which the authors analyze the factors associated with self-monitoring of physical activity, particularly via an activity monitor, in a population with inflammatory arthritis from 4 countries. Major comments: Given the proposed design, it does not seem reasonable to conclude on the impact of the devices on adherence to physical activity recommendations. Yet the results of this analysis appear as key messages in the summary. Abstract L51: “There is good use of technology among people with IJDs to measure their own PA signaling the value of such devices in improving overall PA awareness and engagement. PA self-monitoring was associated with meeting public PA health guidelines in a real-life setting.” It’s difficult to understand how the authors jump to this conclusion based on the results. Moreover, this conclusion seems overly supportive of activity trackers based on the results. As a clinician, I find the most relevant results of this study regarding the objectives are: • Only 112 (8%) of the participants used simple body worn devices and 56 (4%) used complex body worn devices regularly.• Of those using simple and complex devices only 6% and 4% respectively received instructions (from a HP) on how to use the
---

	devices (...) but 50%, reported they were instructed how to use questionnaires to measure PA  • Participants confidence/familiarity in using self-reports and devices based physical: both simple and complex sensor reach very low confidence (from 1 to 3 out of 10). I invite the authors to revise the conclusion of this article. Minor comments: Authors affiliations: "10 Health Service Executive, University Hospitals Limerick Group, Limerick". Please add country. Abstract / Objectives and Introduction L 50: "Self-monitoring of (PA) has the potential to track PA behaviour". Self-monitoring is by definition behavioural monitoring. It might have the potential to change behaviour, for example, but to say that it has the potential to monitor PA is for me a pleonasm. Abstract L33: "receiving instruction how to use PA measures". Is it an outcome measure? Please adjust. Abstract L56: "PA self-monitoring was associated with meeting public PA health guidelines in a real-life setting". Without the use of longitudinal design, this message can't be really informative except for saying again that people with good PA habits are more willing to self-monitor their PA. Introduction P5 L3 : « While self-reported measures such as diaries and questionnaires as well as device-based PA measures can raise awareness and motivate patients to initiate and maintain PA, especially device-based measures show potential to mirror actual PA behaviour (14). »: This sentence can be rewritten for clarity. Methods L5 P6 : « This was organized following the same structure in all countries to identify missing or problematic questions/constructs in the questionnaire." Please explain. Result Table 2: I wonder if the column "Comparison between countries with corresponding p-value" is informative. Results, paragraph "Perceived importance and confidence in self-monitoring of PA: The results on the source of PA information suggest that patients received PA information from a single source, which is probably not the case. Perhaps the authors could clarify that patients reported the main source if this is the case? Result p13 l60: "not experiencing pain in the arm were less likely to fulfil the recommendations of PA". This seems contradictory and should perhaps be developed in the discussion. Discussion P17 l24 "it clearly highlights that using PA devices is one of the contributors to higher/increased PA in this and other populations". I recommend that the authors be more nuanced given the cross-sectional design used in this study.
--	---

REVIEWER	Backhouse, Michael University of Warwick Warwick Medical School, Warwick Clinical Trials Unit
REVIEW RETURNED	09-Sep-2022

GENERAL COMMENTS	Thank you for submitting this paper which explores an important topic for health professionals and people with IJD.
---

	Although the paper was well written, my main concern was that the survey was conducted between 2015 & 2016. That means some of the responses will now be 7 years old but this is not mentioned in the paper. Patient behaviour is likely to have changed greatly in that time, particularly around the use of technology and activity monitors which seem much more common now. Although in my opinion, this does not change the quality of the data, it does change it's relevance to current practice and it is important that this is appropriately reflected in all aspects of the manuscript. In particular this needs to be made clear in the abstract, discussion, conclusion, and potentially the title. Other specific points are identified below: Abstract P3 L33 – consider changing 'receiving' to 'receipt of' P3 L45 & 46 – The terms 'complex' and 'simple' devices are not standard so need defining Strengths & Limitations P4 L7 – I presume 'ipeople' is a typo rather than a new innovation from Apple? Introduction P4 L26 – Please consider adding 'achieving' before 'sufficient PA' P4 L56 – please consider adding a comment on how well evaluated these measurement properties are P5 L4 – this sentence is v long and would benefit from being clearer. What do you mean by 'mirror actual PA behaviour' P5 L37 – please change 'contribute to' to 'associated with' to avoid inferring causality P6 L47 – please check with the journal guidelines to ensure adequate information is provided on each ethical approval. Table 1 Please correct typo on age criteria How were educational level defined and was this valid between different countries Please be consistent with terminology between the use of biologic DMARDs and Synthetic DMARDs Table 2 I may have missed it but if the information on how the p values were calculated is not included, please add it to the manuscript. There appears to be very large variation between countries in almost all parameters reported. This raises questions about how valid it is to combine it in the data in the regression modelling. Both of these points, as well as reasons why this happened and implications, warrant discussion in the discussion section. Results P13 L10 – you highlight that respondents have a higher level of confidence in using certain techniques. Please expand upon this and it's implications for practice in the discussion (with the caveat that this data is now 6 to 7 years old). Discussion P17 L26 – the previous sentence correctly states that you cannot infer causality from a cross sectional study, but then this sentence you state that PA devices contribute to increased PA. Please
--	--

	clarify in the text that this inference is made from other studies, rather than this one. Having looked at the questionnaire included in the supplementary material, I wonder if you have sought to validate some of the responses from patients around the use of specific types of measure? For example, how well do you think people distinguish between a pedometer and accelerometer based device, the term 'accelerometer' is probably not well known by the general population (and the image includes a mobile phone), plus why weren't smart watches specified? I'm sure these issues would have led to some confusion.
--	--

VERSION 1 – AUTHOR RESPONSE

Reviewer 1	Given the proposed design, it does not seem reasonable to conclude on the impact of the devices on adherence to physical activity recommendations. Yet the results of this analysis appear as key messages in the summary. Abstract L51: "There is good use of technology among people with IJDs to measure their own PA signaling the value of such devices in improving overall PA awareness and engagement. PA self-monitoring was associated with meeting public PA health guidelines in a real-life setting." It's difficult to understand how the authors jump to this conclusion based on the results. Moreover, this conclusion seems overly supportive of activity trackers based on the results. As a clinician, I find the most relevant results of this	We acknowledge your comments and we have now revised our conclusions based on your feedback and brought it up in the discussion. Abstract, page 4, lines 4-8 Discussion, page 17, line 25, page 18, lines 1-4 and 16-20, page 20, lines 12-17
-------------------	---	--

	study regarding the objectives are:  • Only 112 (8%) of the participants used simple body worn devices and 56 (4%) used complex body worn devices regularly. • Of those using simple and complex devices only 6% and 4% respectively received instructions (from a HP) on how to use the devices (...) but 50%, reported they were instructed how to use questionnaires to measure PA 	
--	--	--

	 • Participants confidence/familiarity in using self-reports and devices based physical: both simple and complex sensor reach very low confidence (from 1 to 3 out of 10). I invite the authors to revise the conclusion of this article.	
Reviewer 1	Authors affiliations: “10 Health Service Executive, University Hospitals Limerick Group, Limerick”. Please add country.	Thank you for highlighting this mistake. Ireland has been added Page 1,, line 23
Reviewer 1	Abstract / Objectives and Introduction L 50: “Self-monitoring of (PA) has the potential to track PA behaviour”. Self-monitoring is by definition behavioural monitoring. It might have the potential to change behaviour, for example, but to say that it has the potential to monitor PA is for me a pleonasm.	Thank you for this input. We agree that self-monitoring of physical activity might contribute in changing behaviour and increase physical activity levels rather than only monitoring physical activity. We have rephrased the sentence in the abstract and in the introduction. Abstract, Page 3, line 3

		Introduction, Page 5, line 7
Reviewer 1	Abstract L33: “receiving instructions how to use PA measures”. Is it an outcome measure? Please adjust.	Thank you for pointing out the need for adjustment. “Receiving instruction” from a health professional is an outcome measure and we have added additional information in the method section/questionnaire. There was not room in the abstract to add this due to word restrictions, we hope that it is sufficient to change only in the methods. Methods, page 7, lines 22-23
Reviewer 1	Abstract L56: “PA self-monitoring was associated with meeting public PA health guidelines in a real-life setting”. Without the use of longitudinal design, this message can’t be really informative except for saying again that people with good PA habits are more willing to self-monitor their PA.	We acknowledge your input and have revised the conclusion in the abstract and in the discussion. Abstract, page 4, lines 4-8 Discussion, page 21, lines 10-11
Reviewer 1	Introduction P5 L3 : « While self-reported measures such as diaries and questionnaires as well as device-based PA measures can raise awareness and motivate patients to initiate and maintain PA, especially device-based measures	Thank you for pointing out the need for clarifying the sentence. We have now rephrased this in the introduction: Introduction, page5, lines 12-17

	show potential to mirror actual PA behaviour (14). »: This sentence can be rewritten for clarity.	
--	---	--

Reviewer 1	Methods L5 P6 : « This was organized following the same structure in all countries to identify missing or problematic questions/constructs in the questionnaire.” Please explain.	We understand that this was not very well explained and have revised it to be clearer. Methods, page 6, lines 17-21 In the “Questionnaire” section it is also explained and we have also added information there, page 8, lines 2-15
Reviewer 1	Result Table 2: I wonder if the column "Comparison between countries with corresponding p-value" is informative.	Thank you for this input. Based on this comment we have decided to use descriptive statistics to present the demographic, disease-related factors and fulfilment of physical activity recommendations (see Table 2). We have thus deleted the p-values in Table 2 and in the manuscript.
Reviewer 1	Results, paragraph “Perceived importance and confidence in self-monitoring of PA: The results on the source of PA information suggest that patients received PA information from a single source, which is probably not the case. Perhaps the authors could clarify that patients reported the main source if this is the case?	Thank you for this valuable comment. We have clarified this in the result section. Results, page 13, line 20
Reviewer 1	Result p13 l60: “not experiencing pain in the arm were less likely to fulfil the recommendations of PA”. This seems contradictory and should perhaps be developed in the	Thank you for this comment. Based on comments from reviewer 2, we have decided to include “country” as an independent variable in the regression model in order to

	discussion.	control for differences between the countries. In the revised results, significant factors associated with the fulfilment of physical activity recommendations in the multiple regression were: 1. country, 2. no perceived activity limitations and 3. use of devices to measure physical activity. Thus we have instead focused on discussing these results. Results, page 14, lines 15-19, page 16, lines 2-10 Table 4, page15-16 Discussion, page 18, lines 22-25, page 19, lines1-17
--	--------------------	---

Reviewer 1	Discussion P17 l24 “it clearly highlights that using PA devices is one of the contributors to higher/increased PA in this and other populations”. I recommend that the authors be more nuanced given the cross-sectional design used in this study.	We acknowledge your feedback and have rephrased this part in the discussion. Discussion, page 19, lines 8-11
Reviewer 2	Although the paper was well written, my main concern was that the survey was conducted between 2015 & 2016. That means some of the responses will now be 7 years old but this is not mentioned in the paper. Patient behaviour is likely to have changed greatly in that time, particularly around the use of technology and activity monitors which seem much more common now. Although in my opinion, this does not change	Thank you for this valuable comment. We acknowledge your input and have clarified this in the abstract and in several parts of the discussion. Abstract, page 3, line 11 Discussion, page 17, line 4 and 25, page 18 , lines 1-4, page 19, lines 8-11, page 20, lines 12-17

	the quality of the data, it does change it's relevance to current practice and it is important that this is appropriately reflected in all aspects of the manuscript. In particular this needs to be made clear in the abstract, discussion, conclusion, and potentially the title.	
Reviewer 2	Abstract P3 L33 – consider changing 'receiving' to 'receipt of'	Thank you. We have revised accordingly. Abstract, page 3, line 15
Reviewer 2	Abstract P3 L45 & 46 – The terms 'complex' and 'simple' devices are not standard so need defining	Thank you, we acknowledge your comment! We are aware that the terms "complex" and "simple" devices are not standard. We have considered changing to the more common term today; "wearable tracker", but we still need an explanation/definition on how to distinguish between simple and complex devices. Due to word restrictions it's difficult to add a definition in the abstract and we hope that the definition given in the method section is enough.
Reviewer 2	Introduction P4 L26 – Please consider adding 'achieving' before 'sufficient PA'	Thank you for this comment. We agree and have added "achieving".

		Introduction, page 4, line 23
--	--	-------------------------------

Reviewer 2	Introduction P4 L56 – please consider adding a comment on how well evaluated these measurement properties are	Thank you for this suggestion. We have revised our short summary on the current evidence on the measurement properties and use of PA tools in patients with arthritis in our introduction. Introduction, page 5, lines 17-23
Reviewer 2	Introduction P5 L4 – this sentence is v long and would benefit from being clearer. What do you mean by ‘mirror actual PA behaviour’	Thank you for pointing out the need for clarifying the sentence. We have rephrased this in the Introduction. Introduction, page 5, lines 15-20
Reviewer 2	Introduction P5 L37 – please change ‘contribute to’ to ‘associated with’ to avoid inferring causality	We acknowledge this input and have changed contribute to associated with, in the abstract and in the introduction and we have also changed the title. Title, page 1, lines 1-4 Abstract, page 3 lines 5-10 Introduction, page 6, lines 8-10
Reviewer 2	Introduction P6 L47 – please check with the journal guidelines to ensure adequate information is provided on each ethical approval.	Thank you for this input. The BMJ Opens information states that a formal ethical approval is required. We have obtained ethical approvals by each participating country’s relevant research ethics committees and hope this is correct. We have been in contact with the journal and the Editor will look into it once the revision is done.
Reviewer 2	Table 1	Thank you for bringing these points to our attention

	Please correct typo on age criteria How were educational level defined and was this valid between different countries.	The typo on age criteria is corrected in Table 1. Primary and secondary levels were defined as basic education and the third level as higher. The categorization of the levels of education between countries was valid. We have clarified this in Table 1, 2 and 4 Biologic DMARDs and synthetic DMARDs have been revised in Table 2.
--	--	---

	Please be consistent with terminology between the use of biologic DMARDs and Synthetic DMARDs	
Reviewer 2	Table 2 I may have missed it but if the information on how the p values were calculated is not included, please add it to the manuscript. There appears to be very large variation between countries in almost all parameters reported. This raises questions about how valid it is to combine it in the data in the regression modelling. Both of these points, as well as reasons why this happened and implications, warrant discussion in the discussion section.	Thank you for raising this valid concern. Based on input from reviewer 1, we have decided to use descriptive statistics to present the demographic, disease-related factors and fulfilment of physical activity recommendations (please see revised Table 2). We have therefore deleted the p-values in Table 2. We acknowledge the large variation between countries regarding demographic, disease-related factors and fulfilment of physical activity recommendations.

		We have decided to include “country” as an independent variable in the regression model in order to control for differences between the countries. In the revised results, significant factors associated with the fulfilment of physical activity recommendations in the multiple regression were: 1. country, 2. no perceived activity limitations and 3. use of devices to measure physical activity. Results, page 14, lines 15-19, page 16, lines 2-10 Table 4, page15-16 The differences between countries and factors associated with fulfilment of physical activity recommendations are now discussed in the manuscript. Discussion, page 18, lines 22-25, page 19, lines1-17
--	--	---

Reviewer 2	Results P13 L10 – you highlight that respondents have a higher level of confidence in using certain techniques. Please expand upon this and it’s implications for practice in the discussion (with the caveat that this data is now 6 to 7 years old).	We have expanded the discussion regarding confidence in using certain techniques. Discussion, page 18, lines 16-20
Reviewer 2	Discussion P17 L26 – the previous sentence correctly states that you cannot infer causality from a cross sectional study, but then	We acknowledge your comment, we have now revised the discussion.

	this sentence you state that PA devices contribute to increased PA. Please clarify in the text that this inference is made from other studies, rather than this one.	Discussion, page 19, lines 8-11
Reviewer 2	Discussion Having looked at the questionnaire included in the supplementary material, I wonder if you have sought to validate some of the responses from patients around the use of specific types of measure? For example, how well do you think people distinguish between a pedometer and accelerometer based device, the term 'accelerometer' is probably not well known by the general population (and the image includes a mobile phone), plus why weren't smart watches specified? I'm sure these issues would have led to some confusion.	A type of face-validation was made with four patients from each country. This was made to make sure that patients did understand all questions and could distinguish between the different devices. This is addressed in both the section "Patient and Public involvement" and in the "Questionnaire" section. In the online questionnaire it was possible for the respondents to get additional information about the different devices by clicking on a drop-down menu, explaining for example what an accelerometer, smart phone app or pedometer is. Page 6, lines 19-24, page 8, lines 8-17

VERSION 2 – REVIEW

REVIEWER	Davergne, Thomas Sorbonne Université, Institut Pierre Louis d'Epidémiologie
REVIEW RETURNED	11-Dec-2022
GENERAL COMMENTS	Dear Editor, thank you for giving me the opportunity to review again the manuscript "Self-report and device-based physical activity measures and adherence to physical activity recommendations. A cross-sectional survey among people with inflammatory joint disease in four European countries".

	This study aims to describe the use and knowledge of self-report- and device-based PA measures in people with IJDs in four European countries, and to explore potential determinant of PA. All comments have been addressed. Here are the remaining minor comments. P3 L33: "1, 305", there is an extra space? P3 L54: What is a good use of technology? The authors could be more precise regarding this point.
REVIEWER	Backhouse, Michael University of Warwick Warwick Medical School, Warwick Clinical Trials Unit
REVIEW RETURNED	21-Nov-2022
GENERAL COMMENTS	Thank you for revising your manuscript and considering my previous comments. I have no further comments

VERSION 2 – AUTHOR RESPONSE

For some reason the track changes is not visible in the abstract section of the uploaded manuscript, "Main document – marked copy", although I uploaded the document with the changes visible. In the Conclusion in the main manuscript, the changes are visible. I have tried to upload the document several times but it turned out to be the same every time I did this.

I am sorry for this inconvenience and I have tried to make the corrections as clear as possible below.

Page 3, line 33. Thank you for pointing out the extra space which is corrected now (1,305)

Page 3, line 54. We agree that the sentence "What is a good use of technology" could be better expressed and have changed this to: "Almost half of the participants with IJDs used self- or device based PA measures,"

Page 48, line 12. In the Conclusion section we have changed the sentence: "The study found that there is good use of technology among people with IJDs to measure their own PA, signaling the value of such devices in improving overall PA awareness and engagement", into "The study found that almost half of the participants with IJDs used self- or device based PA measures, signaling the value of such devices in improving overall PA awareness and engagement".